# Longtriever: a Pre-trained Long Text Encoder for Dense Document Retrieval

**Junhan Yang**[*]
USTC
Hefei, China
yangjun2@mail.ustc.edu.cn

**Zheng Liu**[†]
MSRA
Beijing, China
zhengliu@microsoft.com

**Chaozhuo Li**
MSRA
Beijing, China
cli@microsoft.com

**Guangzhong Sun**
USTC
Hefei, China
gzsun@ustc.edu.cn

**Xing Xie**
MSRA
Beijing, China
xingx@microsoft.com

## Abstract

Pre-trained language models (PLMs) have achieved the preeminent position in dense retrieval due to their powerful capacity in modeling intrinsic semantics. However, most existing PLM-based retrieval models encounter substantial computational costs and are infeasible for processing long documents. In this paper, a novel retrieval model Longtriever is proposed to embrace three core challenges of long document retrieval: substantial computational cost, incomprehensive document understanding, and scarce annotations. Longtriever splits long documents into short blocks and then efficiently models the local semantics within a block and the global context semantics across blocks in a tightly-coupled manner. A pre-training phase is further proposed to empower Longtriever to achieve a better understanding of underlying semantic correlations. Experimental results on two popular benchmark datasets demonstrate the superiority of our proposal. The source code is released at https://github.com/SamuelYang1/Longtriever .

## 1 Introduction

Document retrieval aims at retrieving the most relevant documents from a vast corpus in response to an input query (Guo et al., 2022), facilitating a myriad of applications such as web search (Xiao et al., 2022a) and question answering (Karpukhin et al., 2020). Recent works (Karpukhin et al., 2020; Xiong et al., 2020) generally first embed queries and documents into low-dimensional dense vectors, and then subsequently calculate their relevance based on these vectors, dubbed as dense retrieval. Due to their powerful capacity in modeling

the intrinsic semantics, pre-trained language models (e.g., BERT (Devlin et al., 2018), RoBERTa (Liu et al., 2019) and DeBERTa (He et al., 2020)) have attained a preeminent position in dense retrieval. Due to the high computational complexity (Vaswani et al., 2017; Lepikhin et al., 2020), most existing PLM-based retrieval models (Gao and Callan, 2021a,b; Liu and Shao, 2022) are designed for passage retrieval (passages are short texts generally no longer than 100 words (Karpukhin et al., 2020)). Nevertheless, long documents are ubiquitous in real life. For example, the average number of words and tokens in the documents of the MS MARCO dataset (Nguyen et al., 2016) are 1,165.46 and 1,631.30 respectively, significantly larger than the maximum input length (e.g. 512) of passage retrieval. In this paper, we aim to investigate the crucial task of long document retrieval, which is challenging due to the following three reasons.

**(1) Substantial computational cost.** One straightforward solution is to directly employ short passage retrievers on long documents. Such approaches suffer from the rapidly growing computational costs due to the quadratic time complexity of vanilla transformers ($O(L^2d)$, where $L$ is the input sequence length, and $d$ denotes the dimension of latent embeddings). Truncating the documents into short passages is a common workaround, which may lead to the potential information loss (Karpukhin et al., 2020; Zhan et al., 2021b).

**(2) Incomprehensive document understanding.** Another typical method is to employ efficient transformers, such as sparse attention transformers (Child et al., 2019; Beltagy et al., 2020; Zaheer et al., 2020) and hierarchical transformers (Zhang et al., 2019; Lin et al., 2021; Wu et al., 2021; Tian et al., 2023). Such approaches contribute to reduc-

---
[*]Work was done during Junhan's internship in MSRA
[†]Corresponding author

ing the computational cost while suffering from severe incomprehensive document understanding. For instance, sparse attention transformers sparsify the full attention via masking, in which several virtual tokens are proposed as the global tokens attended by all tokens (Beltagy et al., 2020; Zaheer et al., 2020). After aggregating the information from all tokens, these global tokens tend to be overloaded, leading to the information mess (Child et al., 2019; Li et al., 2022). In addition, the popular heuristic masking strategy (e.g., random masking) may further aggravate the risk of information loss. The hierarchical transformers usually first split the document into short blocks. The semantics of different blocks are modeled independently, which are further fed into a readout layer in a cascaded manner. Semantics from different blocks are loosely coupled as a token can only attend to the other tokens in the same block and rich long-range context information is largely ignored. Such a loosely-coupled paradigm might be insufficient in modeling the sophisticated cross-block relations and thus cannot learn the comprehensive document representations.

**(3) Scarce annotations.** Existing dense retrieval models generally rely on annotations to fine-tune the PLMs. The scarcity of annotations is further exacerbated in long document retrieval. Compared to short passage retrieval, more training signals are indispensable to achieve an accurate understanding of long documents. However, manually labeled annotations are usually expensive and time-consuming. Hence, the elaborate unsupervised training signals are expected to facilitate the modeling of long documents and alleviate the challenge of annotation scarcity in the fine-tuning phase.

In this paper, we propose a novel dense retrieval model for long documents, dubbed as Longtriever. Longtriever follows the hierarchical paradigm, in which the long document is split into multiple short blocks to ensure the model's efficiency. Each layer of Longtriever consists of two core modules: the intra-block encoder to convey messages among tokens belonging to the same block, and the inter-block encoder to pass information across different blocks. After iteratively stacking multiple Longtriever layers, the tokens in a block are capable of attending to tokens in other blocks, resulting in a tightly-coupled paradigm. Longtriever models the local semantics within a single block and the global context correlations across multiple blocks

simultaneously under a desirable time complexity. Different from previous fine-tuning based methods (Wu et al., 2021; Li et al., 2020; Xiong et al., 2020), here we further design a pre-training phase to gain a better understanding of inherent semantics within the long documents. A novel pre-training task, local masked autoencoder (LMAE), is proposed to mine the intrinsic semantics by reconstructing the raw input of each block based on the global document representations, the block representations, and the original context tokens. The pre-training phase empowers Longtriever with the capability of capturing the unsupervised semantic correlations and contributes to alleviating the reliance on the annotations. Longtriever is extensively evaluated over two popular benchmark datasets, and the experimental results demonstrate the superiority of our proposal.

## 2 Related Work

### 2.1 Dense Retrieval

In the field of dense retrieval models, a bi-encoder architecture is typically employed to separately encode queries and documents, thereby improving search efficiency. This architectural design has found its use in a multitude of applications, such as search engines (Karpukhin et al., 2020), advertising (Lu et al., 2020), and recommendation systems (Xiao et al., 2022b). Karpukhin et al. (2020) illustrated that incorporating in-batch negatives during training could substantially boost the performance of dense retrieval models compared to the traditional BM25 model. Consequently, Xiong et al. (2020) introduced ANCE, proposing the utilization of approximate nearest neighbors as negative samples. To refine the selection of negative samples, Qu et al. (2020) presented RocketQA, which used a more precise cross-encoder. Zhang et al. (2021a) demonstrated the AR2 model, which concurrently trained the bi-encoder and cross-encoder.

Advancements in the quality of dense retrieval are largely attributed to the recent progress in pre-trained language models (Karpukhin et al., 2020; Luan et al., 2021). A common approach entails deploying a universally pre-trained model as the bi-encoder. These models are usually pre-trained using one or multiple masked language modeling (MLM) tasks. For instance, BERT (Devlin et al., 2018) and RoBERTa (Liu et al., 2019) predict masked tokens based on their context. In Ernie (Sun et al., 2019) and Spanbert (Joshi et al., 2020),

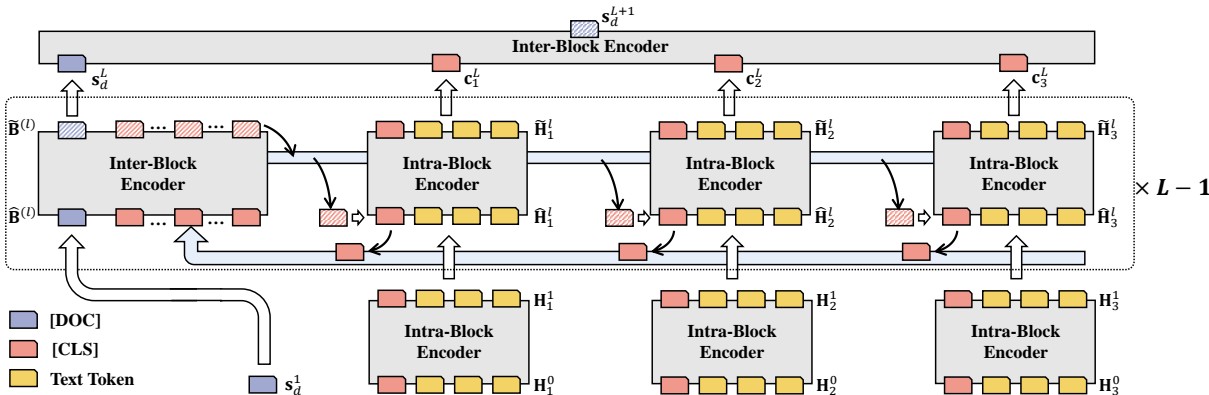

Figure 1: Architecture of Longtriever.

masked entities and spans are predicted, leading to enhanced performance in entity typing and question answering tasks. However, these generic models, usually pre-trained on token-level tasks, might not effectively cultivate the ability to represent sentences (Chang et al., 2020).

To address this limitation, recent studies proposed two primary types of pre-training tasks. The first one is self-contrastive learning (SCL) (Wu et al., 2020; Zhang et al., 2021b; Yan et al., 2021; Liu et al., 2021; Gao et al., 2021), which trains the language models using relevant sentence pairs from an unlabeled corpus. The second is auto-encoding (AE) (Gao and Callan, 2021a; Lu et al., 2021; Liu and Shao, 2022; Pang et al., 2022), which primarily trains language models to reconstruct the input sentence based on the sentence embedding.

## 2.2 Efficient Transformers

Efficient transformers have recently gained substantial attention for their ability to model long documents. Three primary types of efficient transformers have been proposed: sparse attention transformers, hierarchical transformers, and recurrence transformers. Sparse attention transformers, such as Sparse Transformer (Child et al., 2019) and Reformer (Kitaev et al., 2020), sparsify the self-attention matrix to reduce computational costs. Longformer (Beltagy et al., 2020) and BigBird (Zaheer et al., 2020), meanwhile, replace dense attention with a combination of random attention, window attention, and global attention.

Hierarchical transformers, significant for tasks such as document summarization (Zhang et al., 2019) and document ranking (Lin et al., 2021), typically divide a long document into shorter blocks and aggregate these block representations to produce the overall document representation. HIBERT

(Zhang et al., 2019) and Hi-Transformer (Wu et al., 2021) are representative examples of this category.

Recurrence transformers, generally used for generation tasks, also partition the document into short blocks and process them using a recurrence mechanism (Dai et al., 2019). Examples of such models are Transformer-XL and XLNet (Yang et al., 2019), which keep the recurrence mechanism and introduce a permutation language modeling objective to capture bidirectional contextual information. In response to the problem of individual blocks lacking contextual information, ERNIE-Doc (Ding et al., 2020) implements a retrospective feed mechanism, simulating human reading behavior.

## 3 Methodology

### 3.1 Longtriever

Figure 1 illustrates the framework of the proposed Longtriever model. The input long document is tokenized into a sequence of tokens $X = \{x_1, x_2, \cdots, x_L\}$ via WordPiece tokenizer (Wu et al., 2016; Li et al., 2021). This token sequence is further split into a set of blocks $\{T_1, T_2, \cdots, T_N\}$ where $T_i = \{x_{(i-1) \times M+1}, \cdots, x_{i \times M}\}$. $M$ denotes the maximum number of tokens within a single block, which is a pre-defined hyperparameter (e.g., 512). $N = \lceil L/M \rceil$ denotes the number of blocks. Each block is appended with a special token [CLS] as the representation of this entire block. In addition, a special token [DOC] is further padded in the front of the input text as the document representation. Assume $\mathbf{h}_i \in \mathbb{R}^{d \times 1}$ denotes the representation of $i$-th token, $\mathbf{s}_d \in \mathbb{R}^{d \times 1}$ is the document representation, and $\mathbf{c}_i \in \mathbb{R}^{d \times 1}$ represents the [CLS] token of $i$-th block. Each layer of Longtriever consists of two major modules: the inter-block encoder to exchange information across

blocks, and the intra-block encoder to convey messages between tokens within a single block. Next, we will introduce the details of the $l$-th Longtriever layer.

### 3.1.1 Inter-block encoder.

The inter-block encoder aims to depict the global semantics via collecting and dispatching messages for different blocks. The document representation $\mathbf{s}_d^{(l)}$, and the [CLS] tokens $\mathbf{c}_i^{(l)}$ from all blocks are combined into an embedding matrix:

$$\hat{\mathbf{B}}^{(l)} \in \mathbb{R}^{(N+1)\times d} \leftarrow [\mathbf{s}_d^{(l)}, \mathbf{c}_1^{(l)}, \cdots, \mathbf{c}_N^{(l)}]. \quad (1)$$

Matrix $\hat{\mathbf{B}}^{(l)}$ essentially preserves the global semantics of the entire long document. Next, a multi-head transformer is employed on the matrix $\hat{\mathbf{B}}^{(l)}$ to transmit information across different blocks. For an arbitrary attention head, inter-block passing is formalized as:

$$\widetilde{\mathbf{B}}^{(l)} = \text{softmax}\left(\frac{\mathbf{Q}_c^{(l)}\mathbf{K}_c^{(l)\top}}{\sqrt{d}}\right)\mathbf{V}_c^{(l)}, \quad (2)$$

where

$$\begin{cases} \mathbf{Q}_c^{(l)} &= \hat{\mathbf{B}}^{(l)}\mathbf{W}_{Qc}^{(l)}, \\ \mathbf{K}_c^{(l)} &= \hat{\mathbf{B}}^{(l)}\mathbf{W}_{Kc}^{(l)}, \\ \mathbf{V}_c^{(l)} &= \hat{\mathbf{B}}^{(l)}\mathbf{W}_{Vc}^{(l)} \end{cases} \quad (3)$$

in which matrices $\mathbf{W}_{Qc}^{(l)}, \mathbf{W}_{Kc}^{(l)}, \mathbf{W}_{Vc}^{(l)} \in \mathbb{R}^{d\times d}$ are learnable parameters. The information from different blocks is fused and exchanged via the self-attention mechanism of the inter-block encoder, ensuring the [CLS] token of a single block is capable of attending to the global semantics.

### 3.1.2 Intra-block encoder.

The embeddings of [CLS] tokens learned by the inter-block encoder are dispatched back into the corresponding blocks. Given a single block, the embedding matrix fed into the intra-block encoder is formally defined as follows:

$$\hat{\mathbf{H}}^{(l)} \in \mathbb{R}^{(M+1)\times d} \leftarrow [\tilde{\mathbf{c}}_i^{(l)}, \mathbf{h}_1^{(l)}, \cdots, \mathbf{h}_M^{(l)}]. \quad (4)$$

in which $\tilde{\mathbf{c}}_i^{(l)}$ is the [CLS] embedding dispatched from matrix $\widetilde{\mathbf{B}}^{(l)}$ learned by the inter-block encoder. $\mathbf{h}_i^{(l)}$ represents the embedding of the $i$-th token of this block. Similar to the inter-block encoder, a multi-head transformer is applied on the matrix $\hat{\mathbf{H}}^{(l)}$ to conduct token-wise information propagation:

$$\widetilde{\mathbf{H}}^{(l)} = \text{softmax}\left(\frac{\mathbf{Q}_e^{(l)}\mathbf{K}_e^{(l)\top}}{\sqrt{d}}\right)\mathbf{V}_e^{(l)}, \quad (5)$$

where

$$\begin{cases} \mathbf{Q}_e^{(l)} &= \hat{\mathbf{H}}^{(l)}\mathbf{W}_{Qe}^{(l)}, \\ \mathbf{K}_e^{(l)} &= \hat{\mathbf{H}}^{(l)}\mathbf{W}_{Ke}^{(l)}, \\ \mathbf{V}_e^{(l)} &= \hat{\mathbf{H}}^{(l)}\mathbf{W}_{Ve}^{(l)} \end{cases} \quad (6)$$

in which $\mathbf{W}_{Qe}^{(l)}, \mathbf{W}_{Ke}^{(l)}, \mathbf{W}_{Ve}^{(l)} \in \mathbb{R}^{d\times d}$ are learnable matrices. The [CLS] embedding $\tilde{\mathbf{c}}_i^{(l)}$ preserves the global signals from other blocks, which are incorporated to facilitate the modeling of tokens in the belonging block. Different from previous loosely-coupled hierarchical transformers, the modeling of different blocks is tightly coupled together. Each token is capable of attending to tokens in other blocks with the [CLS] token as the intermediary, leading to a comprehensive understanding of the long document semantics. $L$ Longtriever layers are stacked as the model architecture, and the embedding of [DOC] is output as the final document representation.

For the document retrieval task, Longtriever is utilized as both query encoder and document encoder. Following (Karpukhin et al., 2020; Gao and Callan, 2021a; Liu and Shao, 2022), the dot product is selected as the similarity metric. The relevance score between a query and a candidate document is calculated as:

$$rel_{q,d} = LT(q)LT(d)^\top \quad (7)$$

where $q$ is the query, $d$ represents the document and function $LT$ denotes the feed-forward process of Longtriever.

A training sample is defined as $< q, d^+, d_1^-, \cdots, d_n^- >$, in which $d^+$ is the relevant (positive) candidate and $d_i^-$ is the irrelevant (negative) candidate. To ensure the training efficiency, here we adopt the in-batch negative sampling strategy (Karpukhin et al., 2020). The loss function of fine-tuning phase is formalized as the negative log-likelihood of the positive candidates:

$$\mathcal{L}_r = \sum -\log \frac{e^{rel_{q,d^+}}}{e^{rel_{q,d^+}} + \sum_{i=1}^n e^{rel_{q,d_i^-}}}. \quad (8)$$

The time complexity of the inter-block encoder is $O((N+1)^2 d)$. The complexity of a single intra-block encoder is $O((M+1)^2 d)$. Since the input document is split into $N$ blocks, the time cost of intra-block encoders is $O((M+1)^2 * d * N)$. Overall, the time complexity of Longtriever is $O(M^2 Nd + N^2 d)$, which is more efficient than the vanilla transformers $O(M^2 N^2 d)$.

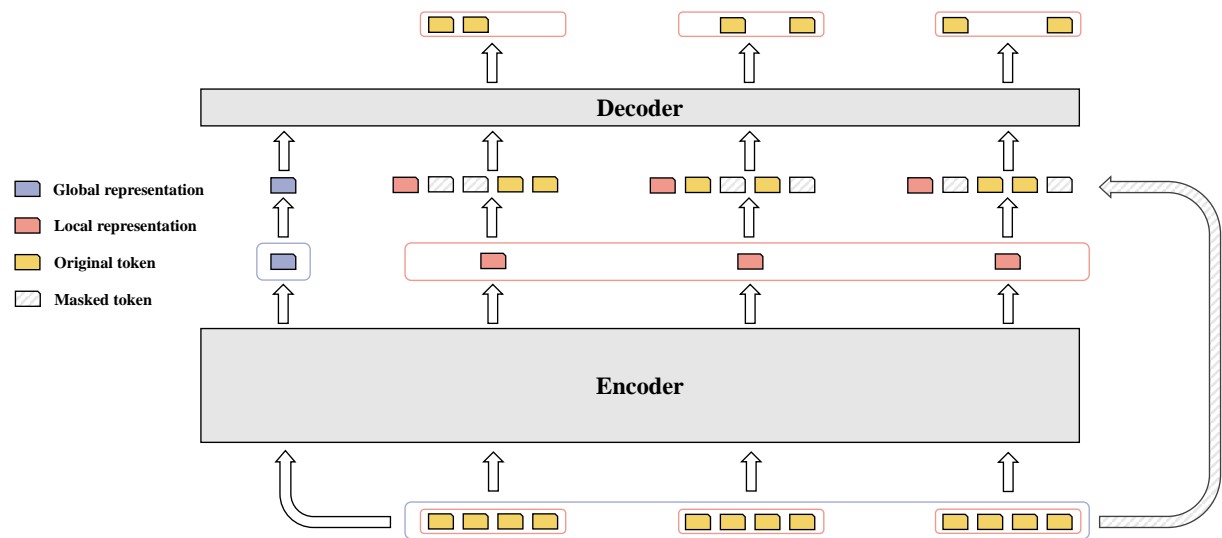

Figure 2: Workflow of LMAE.

## 3.2 Pre-training for long document retrieval

Conventional dense retrieval models solely rely on the annotations to fine-tune PLMs, which is unsuitable for long document retrieval since longer texts generally require more training signals. Thus, here we add a pre-training phase to encode the unsupervised semantic correlations into the proposed Longtriever model.

### 3.2.1 Masked Language Modeling

We adopt the vanilla masked language modeling (MLM) task to pre-train the Longtriever model. The input text $X$ is randomly masked and the objective is defined to predict the masked tokens based on the last hidden states $\mathbf{h}^{(L)}$.

### 3.2.2 Local Masked Autoencoder

To address the annotation scarcity, we propose a novel pre-training task for long document retrieval called local masked autoencoder (LMAE). Shown as figure 2, LMAE first learns two types of representations: the global representation capturing the general semantics of the whole document, and the local representation preserving the specific semantics of the local block. A shallow decoder is further integrated into the Longtriever. LMAE aims to reconstruct the original input tokens via the decoder based on the global and local representations.

In Longtriever, the last hidden state of [DOC], $\mathbf{s}_d^{(L+1)}$ (denoted by $\mathbf{s}_d^{L'}$), is the global representation, and the last hidden states of [CLS] in each block, $\tilde{\mathbf{c}}_i^{(L+1)}$ (denoted by $\tilde{\mathbf{c}}_i^{(L')}$), are the local representations. We utilize a transformer layer as our decoder. For each block $T_i$, we generate distinct query

$\mathbf{H}_i^Q \in \mathbb{R}^{M \times d}$ and key inputs $\mathbf{H}_i^K \in \mathbb{R}^{(M+2) \times d}$ for the multi-head attention layer within the transformer:

$$
\begin{aligned}
\mathbf{H}_i^Q &\leftarrow [\mathbf{s}_d^{(L')} + \mathbf{p}_{(i-1) \times M+1}, ..., \mathbf{s}_d^{(L')} + \mathbf{p}_{i \times M}]; \\
\mathbf{H}_i^K &\leftarrow [\mathbf{s}_d^{(L')}, \tilde{\mathbf{c}}_i^{(L')}, \mathbf{h}_{(i-1) \times M+1}^0, ..., \mathbf{h}_{i \times M}^0].
\end{aligned}
\tag{9}
$$

where $\mathbf{p}$ and $\mathbf{h}^0$ denote the positional embeddings and the orignial token embeddings. After that, a vanilla transformer is employed on the constructed matrices:

$$
\mathbf{H}_i^{\text{Dec}} = \text{softmax} \left( \frac{\mathbf{Q}_i^d \mathbf{K}_i^{d\top}}{\sqrt{d}} + \mathbf{A} \right) \mathbf{V}_i^d,
\tag{10}
$$

where $\mathbf{A}$ is a random mask matrix and

$$
\begin{cases}
\mathbf{Q}_i^d &= \mathbf{H}_i^Q \mathbf{W}_{Qd}, \\
\mathbf{K}_i^d &= \mathbf{H}_i^K \mathbf{W}_{Kd}, \\
\mathbf{V}_i^d &= \mathbf{H}_i^K \mathbf{W}_{Vd}
\end{cases}
\tag{11}
$$

in which $\mathbf{W}_{Qd}, \mathbf{W}_{Kd}, \mathbf{W}_{Vd} \in \mathbb{R}^{d \times d}$ are learnable matrices.

Finally, the output hidden states of the decoder, denoted by $\mathbf{H}_i^{\text{Dec}}$, are processed by the token prediction head $\psi$, and the following objective is optimized:

$$
\mathcal{L}_{\text{LMAE}} = \sum_{x_k \in X} \text{CE}(x_k | \psi(\mathbf{h}_k^{\text{Dec}}))
\tag{12}
$$

where CE is the cross-entropy loss.

| Method | MARCO Dev Doc | | TREC 2019 Doc | |
|---|---|---|---|---|
| | MRR@100 | R@100 | NDCG@10 | R@100 |
| *Sparse retriever* | | | | |
| BM25 (Robertson et al., 2009) | 0.277 | 0.808 | 0.519 | 0.395 |
| DeepCT (Dai and Callan, 2019) | 0.320 | - | 0.544 | - |
| TRECTrad (Craswell et al., 2020) | - | - | 0.549 | - |
| *Dense retriever* | | | | |
| JPQ (Zhan et al., 2021a) | 0.384 | 0.905 | 0.608 | 0.302 |
| RepCONC (Zhan et al., 2022) | 0.399 | 0.911 | 0.600 | 0.305 |
| ANCE (Xiong et al., 2020) | 0.377 | 0.894 | 0.610 | 0.273 |
| STAR (Zhan et al., 2021b) | 0.390 | 0.913 | 0.605 | 0.313 |
| ADORE (Zhan et al., 2021b) | 0.405 | 0.919 | 0.628 | 0.317 |
| BERT (Ma et al., 2022) | 0.389 | 0.877 | 0.594 | 0.301 |
| PROP (Ma et al., 2021a) | 0.394 | 0.884 | 0.596 | 0.298 |
| B-PROP (Ma et al., 2021b) | 0.395 | 0.883 | 0.601 | 0.305 |
| ICT (Lee et al., 2019) | 0.396 | 0.882 | 0.605 | 0.303 |
| SEED (Lu et al., 2021) | 0.396 | 0.902 | 0.605 | 0.307 |
| COSTA (Ma et al., 2022) | 0.422 | 0.919 | 0.626 | 0.320 |
| Longtriever (ours) | **0.434** | **0.940** | **0.645** | **0.356** |

Table 1: Experimental results of retrieval methods.

## 4 Experiments

### 4.1 Experimental Settings

**Datasets**. In order to evaluate the performance of our proposal, extensive experiments are conducted on two popular document retrieval datasets: 1) **MARCO Dev Doc (MS MARCO Document Ranking)** (Nguyen et al., 2016) is a large-scale benchmark dataset for web document retrieval, comprising about 3 million documents, 0.4 million training queries, and 5 thousand development queries. 2) **TREC 2019 Doc (TREC 2019 Document Ranking)** (Craswell et al., 2020) is a test set in MS Marco document ranking task produced by TREC, consisting of 43 queries with more comprehensive labeling. We use the official metrics of these two benchmarks (Nguyen et al., 2016; Craswell et al., 2020). For the MS MARCO document ranking task, we report the mean reciprocal rank at 100 (MRR@100) and recall at 100 (R@100). For the TREC 2019 document ranking task, we report normalized discounted cumulative gain at 10 (NDCG@10) and recall at 100 (R@100). Longtriever is pre-trained on the BookCorpus (Zhu et al., 2015) and English Wikipedia (Devlin et al., 2018).

**Baselines**. Multiple SOTA passage retrieval and document retrieval methods are selected as the baselines. BM25 (Robertson et al., 2009), DeepCT (Dai and Callan, 2019) and TRECTrad (Craswell et al., 2020) are classic sparse document retrievers. JPQ (Zhan et al., 2021a) and RepCONC (Zhan et al., 2022) are recent vector compression methods. ANCE (Xiong et al., 2020), STAR (Zhan et al., 2021b), and ADORE (Zhan et al., 2021b) are complicated fine-tuning methods to enhance dense retrievers.

Several popular PLMs are also introduced as baselines: 1) **BERT** (Devlin et al., 2018) is the most popular pre-trained language model in NLP tasks. 2) **RetroMAE** (Liu and Shao, 2022) is a pre-training paradigm that focuses on dense passage retrieval tasks. It utilizes the masked autoencoder technique and exhibits remarkable performance in these types of tasks. 3) **Parade** (Li et al., 2020) is a method for aggregating passage representations into a document representation. It employs a transformer layer to achieve this aggregation. 4) **Longformer** (Beltagy et al., 2020) uses a technique called "local attention", which allows the model to process much longer sequences than traditional transformer models. 5) **BigBird** (Zaheer et al., 2020) is an efficient transformer model with several sparse attention mechanisms. 6) **Hi-Transformer** (Wu et al., 2021) is a hierarchical interactive transformer for efficient long document modeling. 7) **XLNet** (Yang et al., 2019) is an extension of Transformer-XL (Dai et al., 2019), pre-trained using an autoregressive method to learn bidirectional contexts. It utilizes the recurrent memory mechanism to handle long text.

**Implementation details**. Longtriever employs 24 transformer layers, 12 layers as intra-block en-

| Method | MARCO Dev Doc | | TREC 2019 Doc | |
| --- | --- | --- | --- | --- |
| | MRR@100 | R@100 | NDCG@10 | R@100 |
| *Passage-based Models* | | | | |
| BERT-Passage (Devlin et al., 2018) | 0.296 | 0.859 | 0.522 | 0.285 |
| RetroMAE-Passage (Liu and Shao, 2022) | 0.311 | 0.883 | 0.547 | 0.325 |
| BERT-PARADE (Devlin et al., 2018; Li et al., 2020) | 0.306 | 0.852 | 0.531 | 0.282 |
| RetroMAE-PARADE (Liu and Shao, 2022; Li et al., 2020) | 0.310 | 0.870 | **0.572** | 0.298 |
| *Long-Document Models* | | | | |
| Longformer (Beltagy et al., 2020) | 0.291 | 0.859 | 0.520 | 0.280 |
| BigBird (Zaheer et al., 2020) | 0.293 | 0.859 | 0.544 | 0.281 |
| Hi-Transformer (Wu et al., 2021) | 0.279 | 0.848 | 0.517 | 0.283 |
| XLNet (Yang et al., 2019) | 0.279 | 0.833 | 0.479 | 0.261 |
| Longtriever (ours) | **0.329** | **0.893** | **0.572** | **0.345** |

Table 2: Experimental results of different pre-trained models.

| Method | Time (ms) | Memory (GiB) |
| --- | --- | --- |
| *Passage-based Models* | | |
| BERT-PARADE (4*512) | 674.47 | 3.85 |
| *Long-Document Models* | | |
| BERT-Document (2048) | 1083.98 | 8.42 |
| Longformer (2048) | 970.99 | 6.00 |
| BigBird (2048) | 970.06 | 7.09 |
| Hi-Transformer (4*512) | 674.77 | 4.00 |
| XLNet (4*512) | 2463.49 | 3.99 |
| Longtriever (4*512) | 696.10 | 4.56 |

Table 3: Time and memory costs of different models (batch size is 16).

coders, and 12 layers as inter-block encoders. The dimension of the hidden states is 768, and the vocabulary size is 30,522. The masking ratios for the masked language modeling (MLM) and local masked autoencoder (LMAE) are 30% and 50%. The maximum text length for each block is 512, and the maximum number of blocks is 8. The model is continuously pre-trained from the BERT checkpoint on $8\times$ NVIDIA A100 (40GB) GPUs for 8 epochs with a batch size of 3 (per device), which takes about 3 days. We use the AdamW (Loshchilov and Hutter, 2017) as the optimizer. The peak learning rate is set to 1e-4, with linear warmup over 0.1 ratio and linear decay. The weight decay is set to 0.01.

## 4.2 Main Results

Following previous work (Ma et al., 2022), Longtriever is first fine-tuned on the MS MARCO Passage Ranking dataset and subsequently on the MS MARCO Document Ranking dataset. Each fine-tuning stage includes three iterations: one iteration with BM25 negatives, and two iterations

with hard negatives. Longtriever is trained five times on each dataset, and the average performance is reported. Table 1 presents the retrieval performance of Longtriever and SOTA models. One can clearly see that Longtriever consistently achieves the best performance on all metrics over all datasets. Specifically, Longtriever surpasses the strongest baselines by +2.84% (MRR@100) and +2.29% (Recall@100) on MARCO Dev Doc, and by +2.71% (NDCG@10) and +11.25% (Recall@100) on TREC 2019 Doc. By enjoying the merits of nested inter-block and intra-block aggregations, Longtriever is capable of precisely modeling the semantics within each block, and comprehensively capturing the global semantic correlations between different blocks, leading to superior performance.

In order to further investigate the superiority of Longtriever on long document retrieval, we also present the results of SOTA pre-trained models in Table 2. All models are fine-tuned using in-batch negatives for a single iteration. The notation "*-Passage" indicates that the model takes the first 512 tokens of the text as the input (Ma et al., 2022). The results clearly demonstrate that Longtriever consistently achieves superior performance on both datasets.

The time and memory costs of various models are evaluated on a single NVIDIA V100 GPU with 32GB memory. To ensure a fair comparison, all models are given a batch of 16 documents as input, each document comprised of 2048 tokens or 4 blocks of 512 length. BERT-Passage is modified to BERT-Document for this comparison (The input length is expanded from 512 to 2048). RetroMAE is omitted, as it shares the same backbone with

| No. | Method | MARCO Dev Doc | |
| | | MRR@100 | R@100 |
|---|---|---|---|
| **I.** | Longtriever | **0.329** | **0.893** |
| **II.** | I w/o LMAE | 0.307 | 0.852 |
| **III.** | II w/o MLM | 0.280 | 0.845 |
| **IV.** | III w/o [DOC] token | 0.272 | 0.844 |
| **V.** | IV w/o inter-block encoder | 0.249 | 0.823 |

Table 4: Ablation studies on Longtriever.

BERT. The table 3 shows: 1) The time cost analysis reveals that BERT-PARADE, Hi-Transformer, and Longtriever are the fastest models, taking less than 700ms to process a batch, due to their efficient hierarchical architectures. On the other hand, BERT-Document, Longformer, and BigBird take approximately 1000ms due to their similar attention strategies. XLNet is the slowest model with over 2000ms, due to the low parallelism of its recurrent architecture. 2) The memory cost analysis shows that BERT-PARADE, Hi-Transformer, and XLNet consume no more than 4 GiB. Longtriever is slightly higher, as it concatenates an extra token at the beginning of each block in every intra-block encoder layer. As the attention matrix becomes denser, Longformer, BigBird, and BERT-Document consume more memory in that order. Based on the above analysis, it can be concluded that Longtriever provides an appropriate trade-off between performances and costs.

### 4.3 Ablation Study

Table 4 presents the results of the Longtriever model on the MARCO Dev Doc after removing different components. Various components are gradually removed from Longtriever. In Experiment II, the model is pre-trained with only the MLM task. In Experiment III, the model is fine-tuned without continuous pre-training (just the BERT checkpoint is loaded). In Experiment IV, the [DOC] token is removed, and mean pooling is used to aggregate the [CLS] hidden states of each block to obtain the document representation.

Based on the results of Experiments I and III, the proposed pre-training phase is quite useful, yielding 17.50% and 5.68% improvements on MRR@100 and Recall@100. The scarcity of annotations has been effectively alleviated. Moreover, comparing the results of Experiments I and II, one can see that the LMAE task brings a 7.17% improvement on MRR@100 and 4.81% on Recall@100. It means that Longtriver is able to capture the general semantics of the whole document

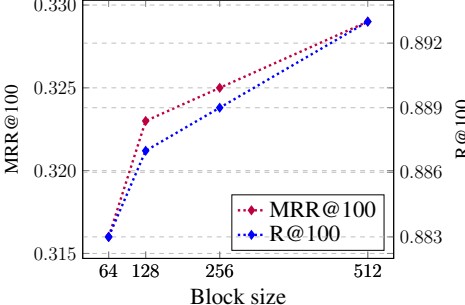

(a) Performance vs. Block size

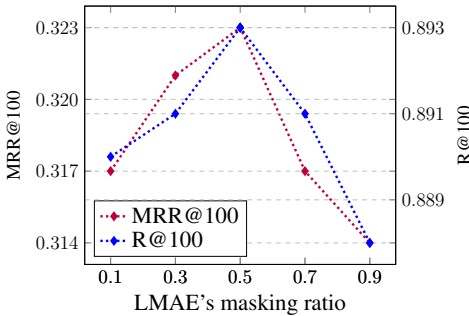

(b) Performance vs. LMAE's masking ratio

Figure 3: (a) and (b) show the performance of Longtriever with varying block sizes and LMAE's masking ratios on MARCO Dev Doc.

through the LMAE task. After removing the [DOC] token, the model's performance presents a decline in Experiment III, which reveals that to generate high-quality document representation, an information collector among blocks is helpful. In addition, the inter-block encoders are also essential for facilitating interactions between different blocks as shown in the comparison between Experiments IV and V. Therefore, integrating inter-block interactions in the modeling of document representation is crucial.

### 4.4 Hyperparameter Study

The influence of block size and LMAE's masking ratio on the performance of Longtriever is depicted in Figure 3. By maintaining a maximum input length of 2,048, the block size is increased from 64 to 512. One can see that model performance consistently increases with larger block sizes. This is reasonable as a larger block size brings more rich intra-block correlations while suffering from inferior model efficiency. We also increase LMAE's masking ratio from 0.1 to 0.9, while MLM's masking ratio is kept at 0.3. The performance improves with LMAE's masking ratio and reaches a peak at

0.5. After that, the performance declines, which demonstrates that a proper masking ratio of LMAE is suitable for Longtriever's pre-training, but a too-aggressive masking ratio harms Longtriever's ability to represent the entire document.

## 5   Conclusion

We propose Longtriever, a novel dense retrieval model for long documents. Longtriever effectively incorporates both local and global semantic modeling, while maintaining a desirable time complexity. Besides, Longtriever is pre-trained with a novel task LMAE to gain a better understanding of inherent semantics within the long documents. Experimental results demonstrate that Longtriever consistently outperforms existing retrieval methods on various document retrieval datasets.

## Limitations

One limitation of our current study is the unexamined performance of the BERT architecture when utilized as a document encoder (BERT-Document). This deficiency is attributed to the substantial GPU memory requirements integral to the fine-tuning process. Our future work endeavours aim to extend the input length of BERT to 2048, which is anticipated to function as a referential upper bound to inform and optimize our methodological approach.

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
