# OpenReview forum: "Longtriever: a Pre-trained Long Text Encoder for Dense Document Retrieval"
_EMNLP/2023/Conference — EMNLP 2023 Main_

### Official Review · Reviewer_fFzZ · 2023-08-03

**Soundness:** 4

**Excitement:**

3: Ambivalent: It has merits (e.g., it reports state-of-the-art results, the idea is nice), but there are key weaknesses (e.g., it describes incremental work), and it can significantly benefit from another round of revision. However, I won't object to accepting it if my co-reviewers champion it.

**Missing References:**

No reference of where the numbers of PROP and B-PROP come from.

**Paper Topic And Main Contributions:**

This paper proposes Longtriever to efficiently model long text through intra-block and inter-block encoders. The main idea is to split text into several sub blocks; then model (or summarize) the information from each sub block using intra-block encoder and use inter-block encoder to further summarize the information from all sub blocks. In addition, the authors propose LMAE pre-training, which is a revised version of MAE pre-training to model long documents, which I think is the main contribution of the paper.

**Questions For The Authors:**

1. Have you evaluated Longtriever’s effectiveness on other NLP tasks of long document used in Longformer, BigBird, Hi-Transformer? It would be nice to see if Longtriever not only outperform other models in retrieval tasks but also other NLP tasks.
2. Where the numbers of PROP and B-PROP come from? I’ve checked the original papers and cannot find the numbers reported in Table 1.

**Reasons To Accept:**

1. The proposed Longtriever architecture is intuitive and the paper is well written and easy to follow.
2. The fine-tuned Longtriever outperforms other existing retrievers on MS MARCO document ranking dataset.

**Reasons To Reject:**

It seems to me that the main contribution of the paper comes from LMAE pre-training, which is a revised version of MAE pre-training specific for modelling long document rather than the proposed architecture. The second issue comes from the evaluation.
1. From the experiments, it seems the main factor for Longtriever to outperform other models comes from LMAE pre-training, which the other models do not experience, rather than the proposed Longtriever architecture (by comparing III in Table 4 with other long-document model in Table 2). It is possible that if we apply the same LMAE pre-training strategy to other models, we can get similar ranking effectiveness.
2. The evaluation is only conducted on MS MARCO document ranking dataset. However, there are many other document ranking datasets, such as Robust04, ClueWeb or Gov2.

**Reproducibility:**

3: Could reproduce the results with some difficulty. The settings of parameters are underspecified or subjectively determined; the training/evaluation data are not widely available.

**Reviewer Confidence:**

4: Quite sure. I tried to check the important points carefully. It's unlikely, though conceivable, that I missed something that should affect my ratings.

---

> ### Author Rebuttal · Authors · 2023-08-29
>
> We are sincerely grateful for the reviewer's recognition and the precious comments towards our work! We would like to take this opportunity to address all the questions within the review comments.
>
> - "From the experiments, it seems the main factor for Longtriever to outperform other models comes from LMAE pre-training, which the other models do not experience, rather than the proposed Longtriever architecture (by comparing III in Table 4 with other long-document models in Table 2). It is possible that if we apply the same LMAE pre-training strategy to other models, we can get similar ranking effectiveness."
>
> It is indeed necessary to compare different model architectures based on the same pre-training recipe. Note that LMAE depends on the model architecture of Longtriever, i.e. intra & inter block encoder, making it non-trivial to be applied to models like BigBird. However, we may find common ground by ablating LMAE and having all the methods pre-trained by MLM (XLNET is the only exception, whose pre-training method GAR was found to be more effective in its original work). Based on this unified pre-training condition, the experiment is performed on MSMARCO-Doc Dev:
>
> | |MRR@100|R@100|
> |--|---|---|
> |Longformer|0.291|0.859|
> |BigBird|0.293|0.859|
> |Hi-Transformer|0.279|0.848|
> |XLNet|0.279|0.833|
> |Longtriever (w. MLM only)|0.307|0.852|
> |Longtriever|0.329|0.893|
>
> It can be observed that although Longtriever's performance drops if LMAE is ablated, i.e. Longtriever (w. MLM only), it still outperforms baselines other baselines like Longformer, BigBird, which follow the same pre-training strategy as Longtriever (w. MLM only). Therefore, the effectiveness of Longtriever's model architecture can be verified. In fact, the corresponding result was included in the original paper but marked as Longtriever "w/o MLM" by mistake. We will correct this mistake in our manuscript.
>
> - “The evaluation is only conducted on MS MARCO document ranking dataset. However, there are many other document ranking datasets, such as Robust04, ClueWeb or Gov2.”
>
> We analyze Longtriever's performance on Robust04 (the other two call for specific applications from the dataset owner, which is not time feasible within the response period). Due to the highly limited time and computation resources, we directly apply our saved checkpoints from the previous experiment.
> We use the saved checkpoints in Table 2 where all methods are briefly fine-tuned with MSMARCO query (without using enhancement strategies, like hard negatives). Same as our results on MSMARCO Doc Dev and TREC'19 Doc, Longtriever maintains a notable advantage over the strongest baselines in our original experiment.
> | |Robust04(NDCG@20)|
> |--|--|
> |RetroMAE-PARADE|0.331|
> |BigBird|0.306|
> |Longtriever|0.388|
>
> We also apply Longtriever's saved checkpoint fine-tuned with hard negatives. For this case, it achieves a NDCG@20 of 0.468, which is much better than the above checkpoint from brief fine-tuning as well as Indri initial rankings' 0.409. However, knowing that the optimization of fine-tuning strategy is not the focus of our work, the relative improvement may already well support the effectiveness of Longtriever.
>
> - "Have you evaluated Longtriever’s effectiveness on other NLP tasks of long document used in Longformer, BigBird, Hi-Transformer? It would be nice to see if Longtriever not only outperform other models in retrieval tasks but also other NLP tasks."
>
> Note that Longtriever is designed mainly for the retrieval of long documents, given its proposed model architecture and pre-training method. Although it can still be applied to deal with short texts (e.g., sentences or passages) on general representation tasks (e.g., STS, clustering, classification, etc.), there is no guarantee of its performance in other applications. In this place, we introduce a brief set of experiments, where Longtriever is compared with SimCSE on STS and SentEval. We may observe that Longretriever's performance is only marginally better than SimCSE. As said, such a result is expected because Longtriever focuses on one specific type of application scenario.
> | |Semantic Textual Similarity| | | | | | | |
> |--|----|---- |---- |---- |---- |--- |--- |--- |
> |Methods|STS12|STS13|STS14|STS15|STS16|STS-B|STS-R|Average|
> |SimCSE-BERT|75.30|84.67|80.19|85.40|80.82|84.25|80.39|81.57|
> |Longretriever|78.28|82.64|79.93|86.03|82.84|84.09|79.08|81.84|
> | |SentEval Transfer Tasks| | | | | | | |
> |Methods|MR|CR|SUBJ|MPQA|SST|TREC|MRPC|Average|
> |SimCSE-BERT|82.69|89.25|94.81|89.59|87.31|88.40|73.51|86.51|
> |Longtriever|81.92|89.19|94.73|90.17|86.33|90.60|75.94|86.98|
>
> - "Where the numbers of PROP and B-PROP come from? I’ve checked the original papers and cannot find the numbers reported in Table 1."
>
> In fact, both results are collected from COSTA [1], which is the most recent work from the same group of authors of PROP/B-PROP. Such a difference is probably because the reported numbers in PROP/B-PROP were results from re-ranking, where the performances of first-stage retrieval cannot be directly observed. Therefore, our cited results of PROP/B-PROP present a fair comparison with other methods (all based on first-stage retrieval).
>
> [1] Pre-train a Discriminative Text Encoder for Dense Retrieval via Contrastive Span Prediction, Ma et. al., SIGIR'22
>
> We will update our paper accordingly based on your suggestions and our newly presented analysis. Thanks again for your precious comments!

---

### Official Review · Reviewer_kLS1 · 2023-08-04

**Soundness:** 4

**Excitement:**

3: Ambivalent: It has merits (e.g., it reports state-of-the-art results, the idea is nice), but there are key weaknesses (e.g., it describes incremental work), and it can significantly benefit from another round of revision. However, I won't object to accepting it if my co-reviewers champion it.

**Missing References:**

[1] Lexicon-Bottlenecked Pretraining for Large-Scale Retrieval. ICLR 2023.\
[2] Fine-Grained Distillation for Long Document Retrieval.

**Paper Topic And Main Contributions:**

This paper presents Longtriever, an innovative model that seamlessly integrates local and global semantics for long text retrieval while maintaining computational efficiency. The authors introduce a novel pre-training task, the Local Masked Autoencoder (LMAE), which enhances the model's ability to capture the underlying meaning of long texts by leveraging both local and global representations.

**Questions For The Authors:**

As mentioned in "Reasons To Reject" above

**Reasons To Accept:**

- The work proposes a novel pre-training task, the Local Masked Autoencoder (LMAE), which enhances the model's ability to capture the underlying meaning of long texts by leveraging both local and global representations.
- The writing and organization of the paper are well, making it easy to follow and understand.

**Reasons To Reject:**

- The proposed LMAE approach bears a strong resemblance to LexMAE [1], yet the paper fails to provide a comparative analysis or discussion of the two methods.
- The experimental results presented in the paper do not include comparisons with other state-of-the-art models, such as LexMAE [1] and FDG [2].
- The description of the experimental setup is unclear, leaving readers uncertain about key experimental details. For instance, it is not specified in Table 1 whether all methods, including the proposed approach, use a sequence length of 512 as input.

[1] Lexicon-Bottlenecked Pretraining for Large-Scale Retrieval. ICLR 2023.\
[2] Fine-Grained Distillation for Long Document Retrieval.

**Reproducibility:**

4: Could mostly reproduce the results, but there may be some variation because of sample variance or minor variations in their interpretation of the protocol or method.

**Reviewer Confidence:**

4: Quite sure. I tried to check the important points carefully. It's unlikely, though conceivable, that I missed something that should affect my ratings.

---

> ### Author Rebuttal · Authors · 2023-08-29
>
> We are sincerely grateful for your recognition of our work and your constructive suggestions. We would like to take this opportunity to address all your questions with the following clarifications.
>
> - "The proposed LMAE approach bears a strong resemblance to LexMAE [1], yet the paper fails to provide a comparative analysis or discussion of the two methods."
>
> The relationship and difference between LMAE and LexMAE are discussed as follows. LMAE resembles LexMAE given the fact that both methods leverage autoencoding to strengthen the representation capability of the document encoder. Particularly, the output embeddings from the document encoder are used to reconstruct the original document on top of a decoding network. In this sense, both works are related to MAE [1], which can be regarded as MAE's derivatives in the text domain. All these methods indicate that the masked autoencoding style pre-training can be highly effective in helping the encoding models generate discriminative embeddings.
>
> [1] Masked Autoencoders Are Scalable Vision Learners
>
> The distinction between LMAE and LexMAE is that the two methods target different representation capabilities of the document encoder, which leads to a completely differentiated pre-training workflow. LMAE is for dense retrieval, where the pre-training targets improving the semantic representation capability for the [CLS]'s embedding. Therefore, the reconstruction is made based on the [CLS]'s embedding. In contrast, LexMAE focuses on sparse retrieval, where the pre-training targets improving the lexical representation capability of the document encoder. To this end, it transforms the entire output embeddings into the vocabulary space and reconstructs the original document on top of it.
>
> - "The experimental results presented in the paper do not include comparisons with other state-of-the-art models, such as LexMAE and FDG."
>
> Despite that LexMAE and FDG are also dedicated to document retrieval, they are two parallel works to Longtriever. LexMAE is about improving the sparse retrieval of documents, and FDG is about leveraging knowledge distillation to improve document retrieval. By comparison, Longtriever is about dense retrieval, where new model architecture and pre-training methods are proposed. That's to say, these methods target different aspects of document retrieval, and it is likely to combine them for a better performance. In this place, we add one simple group of experiments by a collaboration of LMAE and LexMAE (since there is no publicly available checkpoint for the original LexMAE, we use the reproduced model in our experiment). According to the following result, LMAE and LexMAE's ensemble leads to a notable improvement over the performance of each single method, which verifies our point of view.
> | |MARCO Dev Doc| |TREC 2019 Doc| |
> |---|----|---|----|--|
> | |MRR@100|R@100|NDCG@10|R@100|
> |LexMAE|0.444|0.925|-|-|
> |LMAE|0.432|0.942|0.651|0.344|
> |LMAE + LexMAE(our reproduction)|0.445|0.949|0.659|0.391|
>
> - "The description of the experimental setup is unclear, leaving readers uncertain about key experimental details. For instance, it is not specified in Table 1 whether all methods, including the proposed approach, use a sequence length of 512 as input."
>
> The baselines in Table 1 (all based on vanilla BERT) use a sequence length of 512 (by truncating the first 512 tokens of the document) due to the max-length limitation of BERT. The baselines in Table 2 (based on hierarchical or sparse transformers) use a sequence length of 2048, which is the same as Longtriever. The experiment result in Table 1 reflects that Longtriever effectively utilizes the additional context of the document (beyond the first 512 tokens) to achieve a more accurate retrieval performance; while the experiment result in Table 2 indicates that Longtriever makes better utilization of the long context compared with other efficient transformer architectures.
>
> We will update our paper in light of your suggestions and our newly presented analysis. Thanks once again for your invaluable feedback!

---

### Official Review · Reviewer_Yhi7 · 2023-08-10

**Soundness:** 4

**Excitement:**

4: Strong: This paper deepens the understanding of some phenomenon or lowers the barriers to an existing research direction.

**Paper Topic And Main Contributions:**

The authors present Longtriever, an innovative dense retrieval approach that capitalizes on the hierarchical paradigm, leveraging both inter- and intra-mechanisms to deepen the comprehension of document semantics. Additionally, the authors introduce a novel pre-training task, LMAE, designed to enhance the overall generalizability of document inputs. Remarkably, Longtriever surpasses all benchmark methods across two dense retrieval datasets, MS MARCO and TREC.

**Questions For The Authors:**

1. In Section 4.4, the examination of block size effects raises a question about the choice to halt the block size at 512. Is this decision influenced by computational limitations? The trend suggests that expanding the block size could potentially yield performance improvements, given the capacity to include more tokens.

**Reasons To Accept:**

1. The paper is well written, providing a clear motivation and view of the comparative landscape.
2. The goals/challenges of document retrieval (e.g., computational cost, document understanding, scare annotations) are very valuable for readers. It is noteworthy that the authors have effectively tackled each of these aspects within their work.
3. Longtriever introduces a novel framework and architectural design for dense retrieval. While the concept of integrating document knowledge into dense retrieval is not new, the authors innovate by presenting a fresh paradigm for aggregating representations through the utilization of inter-intra mechanisms.
4. The authors provide a (very) comprehensive experimental setting, showing their model is competitive with SOTA methods.


**Reasons To Reject:**

1. The ablation studies shed some light on the importance of the author's choices along the way but do not provide any deep insight into the method's mechanism. I encourage the authors to provide such analysis to help readers to understand the proposed method.
2. While Longtriever presents a novel contribution, it's important to note that the LMAE method may not be entirely novel. The fundamental concept behind LMAE resembles a pre-training task found in autoencoders, such as BART, which involves sentence reconstruction through masking.
3. You don't state you will provide the code for your paper upon acceptance (if you had, and I missed it, please let me know), I believe one could reproduce your results without it, but I (strongly) encourage you to provide it for future researchers.
4. The organization of Section 5 is very hard to follow. Notably, Tables 1 and 2 are positioned within Section 3, making it difficult for readers to follow the flow of experimental results seamlessly.


**Reproducibility:**

2: Would be hard pressed to reproduce the results. The contribution depends on data that are simply not available outside the author's institution or consortium; not enough details are provided.

**Reviewer Confidence:**

4: Quite sure. I tried to check the important points carefully. It's unlikely, though conceivable, that I missed something that should affect my ratings.

---

> ### Author Rebuttal · Authors · 2023-08-29
>
> Thanks a lot for your recognition of our work! We highly value the opportunity to improve our work based on your feedback. In the following part, clarifications are made to address your concerns.
> - “The ablation studies shed some light on the importance of the author's choices along the way but do not provide any deep insight into the method's mechanism. I encourage the authors to provide such analysis to help readers to understand the proposed method.”
>
> The analysis is made for the two components within Longretriever: the model architecture and the pre-training method.
> Firstly, the proposed model architecture, i.e. the joint utilization of Inter-block encoder and Intra-block encoder, helps to comprehensively capture long document semantics in a computationally efficient way.
> The computation cost v.s. the vanilla transformer (namely BERT-Document in Table 3) was evaluated in our paper, where Longtriever achieves a notable reduction in terms of both time cost and memory consumption.
>
> | Method             | Time(ms) | Memory |
> |----------------------|----------|--------|
> | BERT-Document (2048) | 1083.98  | 8.42   |
> | Longretriever(4*512) | 696.10   | 4.56   |
>
> The underlying mechanism is that the vanilla transformer is limited by its O(N^2) time complexity when dealing with long documents. With Longretriever, the long document is partitioned into blocks, and the interaction between different blocks is established by the lightweight inter-block encoder; thus, it achieves an almost O(N) time complexity, which significantly contributes to the working efficiency.
>
> The proposed model architecture is not only efficient but also expressive. The underlying mechanism is that although the long document is partitioned into blocks, it may still preserve the long-range dependency on top of the Inter-block encoder. The Inter-block encoder can be regarded as a "message exchanger" between different blocks, which enables the encoding of one block to be conditioned on the context of the entire document. To verify this point, we evaluate the block-to-block similarity between Longretriever and BERT-PARADE (where each block is independently encoded).
>
> |Method 	  |Average Similarity (cosine)|
> |--------|------|
> |BERT-PARADE  |0.645						  |
> |Longretriever|0.812 						  |
>
> It can be observed that Longretriever results in a much higher block-to-block similarity, indicating that different blocks are more closely related during the encoding process.
>
>
> - "While Longtriever presents a novel contribution, it's important to note that the LMAE method may not be entirely novel. The fundamental concept behind LMAE resembles a pre-training task found in autoencoders, such as BART, which involves sentence reconstruction through masking."
>
> We agree that LMAE resembles BART considering that both of them perform reconstruction for the masked input, i.e. the autoencoding operation. However, LMAE differs from BART in terms of how the autoencoding is conducted. Particularly, BART is mostly for generation tasks; thus, the autoencoding is performed in a Seq2Seq manner. LMAE is for representation tasks (encoding a long document into its embedding); thus, the reconstruction is made purely based on the embedding; in other words, it can be thought to be "embedding-to-sequence". The value of LMAE can be observed from the following empirical gain:
> |Method|MRR@100|
> |----------------------|----------|
> |Longtriever|0.329|
> |Longtriever w.o. LMAE|0.307|
> |Longtriever w.o. LMAE & MLM|0.280|
>
> Longtriever w.o. LMAE indicates that LMAE task is disabled and only MLM is preserved for pre-training (which is more similar to the Seq2Seq as performed in BART). Longtriever w.o. LMAE & MLM means LMAE & MLM are both disabled. The above result shows that the autoencoding is helpful (i.e. by performing MLM only as Longtriever w.o. LMAE). However, the performance can be further improved with the incorporation of both LMAE and MLM (as Longtriever).
>
> - "You don't state you will provide the code for your paper upon acceptance (if you had, and I missed it, please let me know), I believe one could reproduce your results without it, but I (strongly) encourage you to provide it for future researchers."
>
> The source code was submitted in our supplementary material together with the paper. They will also be open to public upon the acceptance. We apologize for making no clear statement about this point in the paper.
>
> - “The organization of Section 5 is very hard to follow. Notably, Tables 1 and 2 are positioned within Section 3, making it difficult for readers to follow the flow of experimental results seamlessly.”
>
> We are sorry for having the experiment results placed in different parts of the paper. The placement of the experiment results will be adjusted for the camera-ready manuscript. Particularly, we will move backward Table 1/2 by one page, and put together Table 3/4 and Figure 3 in the same page. By doing so, all the results will be fully included by the section on experiment.
>
> - "In Section 4.4, the examination of block size effects raises a question about the choice to halt the block size at 512. Is this decision influenced by computational limitations? The trend suggests that expanding the block size could potentially yield performance improvements, given the capacity to include more tokens."
>
> Indeed the performance will probably benefit from the further increasing of block size. However, knowing that our intra-block encoder is based on BERT whose max position embedding is 512, we cannot improve the size of the intra-block encoder any further beyond 512. Besides, just as you mentioned, the increasing block size will considerably add to the computation cost, which also prevents the block size from unlimited growth.
>
> We will update our paper accordingly based on your suggestions and our newly presented analysis. Thanks again for your precious comments!

---

### Meta-Review · Area_Chair_eUEU · 2023-09-26

**Recommendation:** 4

**Metareview:**

This paper presents the Longtriever to efficiently model long text through intra-block and inter-block encoders, and then proposes a pre-trained LMAE task. This paper is well-presented, and the idea is interesting. Longtriever achieves superior performance on two dense retrieval datasets, MS MARCO and TREC. All reviewers were positive about this work, seeing it as solid work falling in an important area for the NLP community.

---

### Decision · Program_Chairs · 2023-10-07

**Decision:**

Accept-Main

**Comment:**

This paper presents the Longtriever to efficiently model long text through intra-block and inter-block encoders, and then proposes a pre-trained LMAE task. This paper is well-presented, and the idea is interesting. Longtriever achieves superior performance on two dense retrieval datasets, MS MARCO and TREC. All reviewers were positive about this work, seeing it as solid work falling in an important area for the NLP community.